# The Beneficial Role of Involvement in Alcoholics Anonymous for Existential and Subjective Well-Being of Alcohol-Dependent Individuals? The Model Verification

**DOI:** 10.3390/ijerph19095173

**Published:** 2022-04-24

**Authors:** Marcin Wnuk

**Affiliations:** Department of Psychology, Adam Mickiewicz University, Szamarzewskiego Street 89, 60-568 Poznań, Poland; marwnu@amu.edu.pl; Tel.: +48-664-934-268

**Keywords:** involvement in Alcoholics Anonymous, hope, meaning in life, existential well-being, subjective well-being

## Abstract

Involvement in Alcoholics Anonymous (AA) is an important psychosocial factor for the recovery of alcohol-dependent individuals. Recent studies have confirmed the beneficial role of involvement in AA for abstinence and reduction in drinking alcohol. Little is known about the mechanism underlying the relationship between involvement in AA and subjective well-being. This study aims to verify whether in a sample of Polish AA participants involvement in AA is indirectly related to subjective well-being through existential well-being consisting of hope and meaning in life. The achieved results have confirmed that involvement in AA is positively related to existential well-being, which in turn positively predicts subjective well-being including life satisfaction as well as positive and negative affect. It was confirmed that AA involvement in self-help groups indirectly via existential well-being is related to subjective well-being. Theoretical and practical implications were discussed.

## 1. Introduction

In Poland, alcohol abuse and addiction cause serious social, economic, and health-related problems. The general rate at which alcohol is consumed each year per capita has been increasing since 1999 [1], as have the numbers of individuals using alcohol in a risky and problematic way. In Poland, there are about 800,000 alcohol-dependent individuals and about 2–2.5 million citizens who abuse alcohol [2]. One method of support for alcohol-dependent individuals in Poland is participation in Alcoholics Anonymous (AA). This method of addressing alcoholism is based on the completion of a spiritual 12-step programme [3] and a philosophy of alcohol addiction as a deadly, chronic, progressive, and multifaceted disease negatively affecting the social, physical, mental, emotional and spiritual aspects of life [4]. The roots of this self-help movement come from the United States and date back to 1935.

In Poland, AA meetings have been organised since 1974 [5]. Currently, there are 2300 meetings per week [6]. In the literature one can find well-established discussions about the effectiveness of this form of therapy for individuals with alcohol dependence [7,8,9]. Some studies have indicated involvement in AA as an effective way of alcohol-dependence recovery [10,11,12,13,14,15].

Little is known about the spiritual mechanism underlying the relationship between involvement in AA and subjective well-being. The aim of this study was to test the beneficial role of existential well-being in the indirect relationship between involvement in AA and the subjective well-being in a sample of individuals with alcohol dependence from Poland. In the literature, there is a lack of research regarding predictors of subjective well-being among participants of self-help groups. Most researchers have focused on abstinence duration or reduced drinking as a recovery indicators. Contrary to most recent studies, subjective well-being was used as a recovery indicator instead of abstinence duration or reduced drinking. It is due to the fact that abstinence is not the same phenomenon as sobriety. Abstinence exclusively refers to the symptoms of the disease, while sobriety is something deeper, manifesting in emotional and mental dimensions, being an expression of sobriety [16]. Existential well-being [17,18] is another construct than subjective well-being [19], but both of them are strictly related. It finds reflections in Seligman’s [20] model of happiness approaches for a meaningful and purposeful life as one of three ways to achieve subjective well-being. Additionally, in Frankl’s conception [21] finding meaning in life facilitates fulfillment and satisfaction.

It means that existential well-being is a significant factor in achieving happiness, operationalized as subjective well-being. Recent research has confirmed that existential well-being aspects, such as a meaning in life and hope as a consequence of spiritual growth, lead to subjective well-being [22,23], consisting of life satisfaction and positive and negative affect [19].

## 2. Review of Literature

### 2.1. Involvement in AA and Recovery from Alcohol Addiction

Involvement in AA results in positive alcohol outcomes for many AA participants.

In a study by Walitzer, Dermen, and Barrick [24] among the patients included a group of directive approach to facilitating AA in comparison to the other two groups. Involvement in AA and meeting attendance were positively related to the percentage of days abstinent.

Greenfield and Tonigan’s research [25] spiritual step work predicted the percentage of days abstinent. In another longitudinal study, the duration in AA correlated positively with abstinence, social functioning, and self-efficacy and was negatively related to depression and drinking problems [26]. In a sample of young adults participating in AA, serving as a sponsor and level of affiliation were associated with lower anxiety [27]. In a McKellar, Stewart, and Humphreys [28] study, one-year-posttreatment levels of AA involvement predicted lower alcohol-related problems at the two-year follow-up.

Some studies have proven that having a spiritual awakening as a result of AA engagement has a positive outcome, such as alcohol and drug abstinence [29], fewer depressive symptoms [30], predicting continuous sobriety [31], as well as being less likely to report a craving for alcohol [30]. In a MATCH project AA attendance, number of steps completed, and identifying oneself as an AA member were identified as predictors of one-year post-treatment drinking outcomes [11].

Having a sponsor is an important part of involvement in AA and of connecting with the recovery process. According to recent studies, having a sponsor was associated with longer abstinence [32,33,34], and lesser psychiatric severity [35].

Some authors have tried to find a mechanism of beneficial influence of AA involvement on the recovery process. In most studies the effectiveness of recovery is measured by abstinence or reduction in rate of using alcohol.

In the longitudinal study conducted among individuals with alcohol dependence [36], AA affiliation reduced drinking and drug use indirectly via self-efficacy, motivation, and active coping.

Recovery mechanisms were explored by using potential mediators between AA attendance and alcohol consumption by Kelly et al. [37]. Among aftercare patients, AA attendance indirectly improved the percentage of days abstinent via positive social self-efficacy, spiritual/religious practices, pro-abstinence social networks, and pro-drinking social networks. In the same group of patients, mediators between AA attendance and reduced drinks per drinking day were positive social self-efficacy, negative affect, spiritual/religious practices, depression, and pro-drinking social networks. Among outpatients, relationships between AA attendance and reduced drinks per drinking day as well as percentage of days abstinent were mediated by pro-abstinence social networks, pro-drinking social networks, and positive social self-efficacy [37]. In Blonigen et al. [38]), longer AA participation duration was related to decreased impulsivity (from baseline to one year), which in turn led to a reduction in alcohol-use problems, improvement in social support, improvement in self-efficacy to resist drinking as well as improvement in emotional discharge coping. In their longitudinal research, Kaskutas et al. [38] proved that involvement in AA both directly and indirectly—through support from other AA participants—influenced lower alcohol consumption and less severe problems. This effect was noticed both at the start and after 30 days of abstinence. Besides social and cognitive mechanisms of change as a result of involvement in AA, the literature also describes a spiritual mechanism of change. According to this mechanism, involvement in AA has a positive, indirect influence on reducing alcohol consumption through spirituality/religiousness [39,40,41,42]. Spirituality and religiousness are similar and overlapping constructs, but spirituality is a wider conception includes religiousness as one form of express spirituality [43]. The spiritual growth of AA participants is based on AA involvement and/or religiousness. These forms can be used for religiously inclined individuals, but for religious skeptics, agnostics, and atheists, involvement in AA as a secular way of developing spirituality is preferred. According to Kurtz and White [44], this first mechanism of spiritual growth called secular spirituality emphasizes the role of secular values, and the second one—religious spirituality, focuses on religious values. In this study, the spiritual mechanism was tested using involvement in AA as a manifestation of secular spirituality, which through existential well-being [17,18] is related to subjective well-being.

It is possible that existential variables (such as meaning in life and hope) as an effect of involvement in AA may lead to subjective well-being. It can be treated as a parallel spiritual mechanism to that revealed in recent studies.

### 2.2. Improved Meaning of Life as a Potential Mechanism of Change Associated with AA Involvement

Meaning in life seems to buffer the influence of the risks of alcohol and drug abuse and is an important factor facilitating recovery from addiction. Use of alcohol and psychoactive substances, as well as involvement in other compulsive behaviours such as surfing the internet, gambling, and sex activity, can be a way to cope with the “existential vacuum” as a state of feeling of meaninglessness, hopelessness, senselessness, boredom, and anhedonia [21], leading to dependency. Researchers have indicated that the problem with finding purpose and meaning in life is related to alcohol use [45], substance use [46], alcohol and drug abuse [47,48,49,50], sedative use [51], smartphone and internet addiction [52,53], gambling [54], as well as substantially higher likelihood of future drug misuse [55]. For example, in a sample of students from Romania, meaning in life protected against drug and sedative use as well as unsafe sex [56].

Some authors have confirmed that involvement in self-help fellowships among individuals with alcohol and drug dependence is an effective way to find meaning in life [57,58,59,60,61,62]. For example, in a longitudinal study conducted among 364 individuals diagnosed with alcohol dependence, participating in AA resulted in a significant increase in their meaning in life over a 30-month period [41]. In a sample of Narcotics Anonymous (NA) members from the US, participation in NA comfort affected with the home group and involvement in NA services predicted their purpose in life [63]. Among AA participants from Great Britain, completion of steps 4 and 5 as well as involvement in AA correlated positively with existential well-being [60].

Much research has emphasised on the significant role of purpose and meaning in life in individuals with alcohol dependence recovery [57,64,65,66,67]. For example, in longitudinal research on outcomes from a MATCH project, higher initial levels of purpose in life and increases in purpose in life over time were related with lower initial levels of temptation to drink and decreases in the temptation to drink over time. Decreases in purpose in life were also significantly associated with greater intensity and frequency of drinking and greater drinking-related consequences at the 15-month follow-up [68].

### 2.3. Improved Hope as a Potential Mechanism of Change Associated with AA Involvement

Many approaches to hope have been presented. Some authors indicate that hope is a character trait [69,70,71,72] or positive emotion [73,74,75]. Others have conceptualised hope from the perspective of positive psychology as a strength of character [76]. Hope-focused intervention is a significant issue in the area of substance abuse counselling [77]. Hope is predictive of entering substance abuse treatment [78] and during therapy of facilitating the ease of the required changes and return to health [79]. Additionally, hope is a factor in reducing the risk of relapse [80].

In the literature there is no research regarding the role of involvement in self-help groups in developing or maintaining hope among individuals with alcohol dependence. The results of one available paper indicated that in a sample of people with dual diagnoses of mental and substance use disorders, involvement in self-help groups led to hope, which influenced health-promoting behaviours [81].

Recent research has confirmed that hope is an important factor for the recovery and well-being of individuals with alcohol dependence [82,83,84,85]. In Gutierrez’s study [64] meaning in life via hope indirectly reduced alcohol and drug use. Mathis et al. [83] have indicated that hope predicted drug abstinence at an eight-month follow-up. Additionally, in a sample of AA participants from Poland and participants in Sex and Love Addicted Individuals (SLAA), hope was positively related to different measures of well-being [86,87].

Among AA individuals, hope measured by the Herth Hope Index correlated positively with evaluation of life and correlated negatively with the level of stress [86]. Among SLAA participants, hope improved the desire to live, passion for life and feeling of happiness on a few recent days, as well as satisfaction with friends, health, perspective on the future, life achievements, ways of spending of free time, and satisfaction with sex [87].

## 3. Material and Methods

### 3.1. Participants

The participants of the study were 70 individuals addicted to alcohol, attending AA meetings in Poland. The subjects gave their consent to take part in the study. The surveys were distributed by the psychologist during AA meetings and collected during the next meeting after being completed at home.

### 3.2. Measures

The following tools were used: the Herth Hope Index (HHI), the Purpose in Life Test (PIL), Positive and Negative Affect Schedule (PANAS), Alcoholics Anonymous Involvement Scale (AAIS) and one-item measures regarding length of being in AA, abstinence duration, quantity of completed steps within a 12-step programme, frequency of being at AA meetings, as well as frequency of being chairman at AA meetings.

Affiliation with AA. The Alcoholics Anonymous Involvement Scale (AAIS) was used to assess lifetime AA attendance [88]. AAI consists of 13 items related to involvement in AA attendance and activities, including considering oneself to be a member of AA, going to 90 meetings in 90 days, celebrating an AA birthday, having and/or being a sponsor, having a spiritual awakening, etc. Participants responded either “Yes” or “No”. Additionally, participants were asked about steps completed in alcohol treatment, about steps “worked”, about quantity of meetings attended in the last year, and total number of meetings ever attended. In the research, questions were asked regarding participation in any alcohol treatment, spiritual awakening as a consequence of involvement in AA, total number of meetings, and number of steps “worked”. Additionally, participants were asked about frequency of being chairman at AA meetings during the last year.

Steps quantity. One item measure was used to verify this. Participants declared how many steps they had completed from the 12-step programme.

Hope. The Herth Hope Index (HHI) consists of 12 items and is a measure of hope. Respondents rate each item on a 4-point Likert scale, from 1—strongly disagree, to 4—strongly agree [89]. The reliability of this measure assessed as Cronbach’s α was 0.97 [90]; its test-retest reliability was 0.91 [89].

Meaning in life. The Purpose in Life Test (PIL) consists of 20 items concerning meaning in life, which subjects respond to by indicating a field on the continuum ranging from 1 to 7, where 7 represents the maximum level of meaning in life and 1 represents the minimum level. The score is computed by adding up the responses to all items. The higher the score, the stronger the satisfaction of the need for meaning in life; the lower the score, the greater the existential frustration. The reliability of this test measured as Pearson’s r coefficient was 0.82; with the Spearman-Brown correction, it was 0.90 [50].

Life satisfaction. The Cantril Ladder [91] is a well-known measure used to verify life satisfaction. It consists of one question in which the respondent evaluates his overall satisfaction with life on a scale from 0 (minimum) to 10 (maximum). It was used from future perspective (for the five years). This tool has satisfied psychometrics properties. Czapiński used this scale during a two month interval, obtaining a reliability score of 0.76 [92]. In another project the coefficient of reliability at a two year interval measured, 0.65 [93].

Positive and negative affect. The Positive and Negative Affects Schedule (PANAS) consists of 10 statements related to positive emotional states and another 10 concerning negative ones. Each question is graded from 1 = a little or none to 5 = very frequently. A higher score on the positive affect dimension indicates greater levels of positive affect, and a higher score on the negative affect dimension reflects a greater level of negative affect. Participants were asked to assess their emotional state based on how often they related to particular questions up to the weekend before the survey. According to studies, the reliability of the scale varied from α = 0.86 to 0.89 for the positive affect and α = 0.84 to 0.85 for the negative affect [94].

### 3.3. Conceptual Model

Given the existing research that links AA involvement and well-being, the current study aimed to understand further how AA involvement is related to subjective well-being among AA participants from Poland. A conceptual model of subjective well-being was tested as an indirect result of AA involvement through an existential well-being latent variable consisting of meaning in life and hope. According to Diener’s concept, subjective well-being refers to cognitive and affective well-being dimensions operationalized as a person’s cognitive and affective evaluations of life [19]. In a study, subjective well-being was the latent variable which consisted of life satisfaction as a cognitive indicator and effective indicators regarding positive and negative emotions (for example excited, guilty, hostile, upset, proud, etc.) experienced over the past week.

Existential well-being is a secular dimension of spiritual well-being [17]. MacDonald’s [18] definition of existential well-being was used as a spirituality dimension as expressed through a sense of meaning and purpose in life, and a perception of self as being competent to successfully cope with the difficulties of life and limitations of human existence. Meaning in life and hope are components of existential well-being, which includes realizing values, having goals, controlling one’s destiny, and finding self-acceptance [95]. Involvement in AA was the latent variable consisting of aggregate responses with dichotomous items from AAIS [88] and, separately, two items regarding frequency of AA attendance in the last year and frequency of being chairman in the last year.

### 3.4. Statistical Analyses

All statistical analyses were conducted using IBM SPSS Statistics software (Version 27.0, Chicago, IL, USA). Structural equation modelling (SEM) was used to investigate the relations between involvement in Anonymous Alcoholics and subjective well-being by exploring the role of existential well-being.

The maximum likelihood method of SEM was used to validate the measurement models and the structural equation models. It was due that all values of skewness were between −2 and +2 and kurtosis between −7 and + 7 (see Table 1), which meant that their distribution is closed to normal distribution [96].

The research model assumed that involvement in AA is indirectly related to subjective well-being through existential well-being.

Structural models were tested by applying path analysis to investigate the relationships among the latent variables such as involvement in AA, existential well-being, and subjective well-being. Models were tested using multiple goodness-of-fit indices, as recommended by Brown [97], including the root mean square error of approximation (RMSEA), the comparative fit index (CFI), and goodness-of-fit index (GFI). For the RMSEA, values less than 0.08, and ideally below 0.05, were used to indicate an adequate and reasonable fit to the data [98]. Values of 0.90 or greater, and ideally above 0.95, were used to indicate good model fits for the CFI [98,99].

The level of NFI should exceed 0.90 [100]. Due to the relatively small sample size, the Bollen-Stine bootstrapping method for 5000 samples resamples, and 95% interval confidence [101]. Both bias-corrected (BC) percentile method for 95% confidence intervals were derived. When the values of upper level (UL) and lower level (LL) does not include a 0, the test statistic is significantly different from zero [102].

## 4. Results

### 4.1. Descriptive Statistics

Statistics of demographic variables are presented in Table 1. Most of the study participants were men with secondary and vocational education. On average they have stayed in abstinence longer than six years and have participated in AA longer than eight years. Their mean quantity of completed steps was also relatively high.

Descriptive statistics of measurable variables are presented in Table 2.

Results of the correlation coefficients are shown in Table 3.

### 4.2. Model Verification

The results of path analysis were presented in Figure 1.

According to the achieved results, all values of fit model indicators have confirmed that the model was well fitted to the data RMSEA = 0.025 (90% CI [0.000, 0.112]), GFI = 0.932, CFI = 0.992, chi^2^ = 19.78, *df* = 19, *p* = 0.409 [CMIN/*df* = 1.041]). The RMSEA value was less than required for ideal fitting 0.05 [98]. The level of CFI exceeded 0.95 [98] and the values of CMIN/*df* statistics, based on the chi-square statistic, were lower than the required standard of—2 or 3 [103]. Additionally using the Bollen–Stine bootstrap method (*p* = 0.423) we confirmed that the model was well fitted to the data.

The standardised direct effect of involvement in AA on existential well-being was statistically significant (CI 95% [0.058; 0.649], beta = 0.375, *p* < 0.05)—the same as the standardised direct effect of existential well-being on subjective well-being (CI 95% [0.740; 0.994], beta = 0.896, *p* < 0.01). The model revealed a statistically significant indirect effect between involvement in AA and subjective well-being through existential well-being (CI 95% [0.073; 0.590], beta = 0.336, *p* < 0.05). This means that greater involvement in AA among individuals with alcohol dependence leads to higher existential well-being, which in turn is related to a higher level of positive affect and anticipated life satisfaction as well as a lower level of negative affect.

## 5. Discussion

The aim of this study was to test the positive role of existential well-being in the indirect relationship between involvement in AA and subjective well-being in a sample of individuals with alcohol dependence from Poland. The obtained results confirmed that involvement in AA is indirectly related to subjective well-being via existential well-being, consisting of meaning in life and hope.

According to recent research, the positive role of involvement in AA in the recovery process of individuals with alcohol dependence has been confirmed [30,31,33,38].

Recent studies have indicated that involvement in AA influenced recovery was mostly measured as abstinence and reduced alcohol use through the social and spiritual change mechanisms [103]. The relationships between involvement in AA and abstinence and alcohol use indicators were mediated through social networks and social support [34,37,38,104,105,106], self-efficacy [36,37], motivation and active coping [36], as well as spirituality/religiousness [40,41,42]. The achieved results have confirmed that another mechanism underlies the relationship between involvement in self-help groups and subjective well-being among the alcohol-addicted exists.

In this spiritual mechanism, a positive role fills existential well-being, which connects AA involvement with subjective well-being. Consistent with recent research, it has been proven that involvement in AA could be a facilitator for finding meaning in life [57,58,59,60,61,62] and hope [81] for alcohol-dependent individuals, which in turn is positively related to well-being [82,83,84,86]. The idea that meaning in life, as well as hope, are important factors in the recovery of individuals with alcohol dependence is present both in the theory and practice of counselling [77,107].

Existing research has confirmed that Frankl’s *tragic optimism* [21] could be used as an explanatory symbol of recovery for individuals with alcohol dependence participating in AA. Representatives of this group have found themselves in tragic situations because of their alcohol dependence, yet are able to transform their circumstances into something meaningful despite pain and suffering, finding that life has meaning and purpose, and fostering and maintaining hope. According to the obtained results, the antecedent of their existential well-being was involvement in AA. This variable explained only 14.06% of the variance of existential well-being, suggesting existence of spiritual [64,108], as well as other sources of meaning in life and hope [109,110]. Existential well-being explained 80.28% of subjective well-being, confirming the crucial role of purpose and meaning in life as well as hope in the process of recovery for alcohol-dependent participants in AA meetings. On the one hand, this could mean that existential well-being and subjective well-being are the same or overlapping constructs, or these are the elements of the other more wider conception of general well-being. On the other hand, existential well-being observed variables, such as meaning in life and hope, were only moderately correlated with observed variables of subjective well-being, such as life satisfaction as well as positive and negative affect. This was the proof that there are other but overlapping constructs or indicators of other variables.

The role of existential well-being in the relationship between AA involvement and subjective well-being can be explained based on the idea of “spiritual transformation” proposed by Neff and MacMaster [111] as well as the instillation of hope. Both of these phenomena have emphasised the relevant role of social learning [112]. According to Neff and MacMaster [111], social learning among Alcoholics Anonymous participants facilitates a “spiritual transformation” that influences their behavioural change. In this spiritual transformation process, some spiritual mechanisms, such as a changing perception of God, increased meaning in life, openness to forgiveness, an increase in self-acceptance, increased use of positive coping mechanisms, and increased integration into social circles are involved. The same function facilitates finding meaning in life, as religion can play with the philosophy of AA and 12-step programmes [3,4]. Like religion, the philosophy of AA and the set of rules functioning in self-help groups can constitute a comprehensive framework for perceiving, understanding, and evaluating experiences, and can organise and orient the behaviours of individuals with alcohol dependence [113,114]. In other words, in individuals for whom AA philosophy is an important value, this importance translates into an attitude in life manifesting itself in a specific perception of reality, the emotional attitude to it, and the behaviour towards it. Additionally, like other religious meaning systems, it can influence the formation of goals for self-regulation [115,116].

Receiving the new philosophy of life and building a stable identity as an alcohol dependent individual provides the chance for change and transformation, finding purpose and meaning in life, and supplies the tools necessary to achieve this goal. Additionally, it gives hope for a better life based on the examples of other sober and contented members of AA groups who personify this hope. The instillation of hope decreases a sense of aloneness and increases self-esteem; the realization that one’s personal experience can be of value to others decreases negative affect and increases the sense of belonging and human connectedness [102].

The role of hope as a result of AA involvement leading to recovery of alcohol-dependent patients can be explained based on the Farran, Wilken, and Popovich [73] model. The first of the four main attributes is called the processes of experiencing; it assumes that an individual with alcohol dependence should accept his disease experiences as part of “being”. This is consistent with the first step of the 12 Step Programme, emphasising powerlessness towards alcohol and lack of control in his use [3].

The attribute of spiritual hope is demonstrated by the premise that something higher exists or believing in something that cannot as of yet be proven [73], and corresponds with the second-step expectation that there is a power greater than the alcoholic that could restore his or her sanity [3] The rational-thinking aspect of hope presumes the existence of aims, e.g., regaining the connection to one’s past, present, and future and in maintaining control over one’s life [73]. This aspect of hope has a reflection in Alcoholics Anonymous’ primary purposes, such as staying sober. Only sobriety allows individuals with alcohol dependence to restore their lives and take control over them [4].

Hope in its relational aspect refers to bonding feeling with others. It seems that this forms the basis for achieving unity at AA. The sharing of common aims, mutual understanding, and trust are fundamental for shaping relationships with others. According to Yalom [79], hope is based on being aware of the similarities of life’s experiences with others. AA’s structure leads to identifying with those other AA members who now demonstrate appropriate behaviour upon which one’s own behaviour and attitudes can be modelled.

The research yields some theoretical and practical implications. First of all, in a sample of individuals with alcohol dependence from Poland participating in AA, the existence of another recently explored spiritual mechanism [29,40,41,42] of indirect impact of involvement in AA on subjective well-being mediated by existential well-being was confirmed. This means that practitioners, therapists, and counsellors should engage patients with an alcohol addiction diagnosis to participate in AA meetings as an effective way to cope with dependence. It is recommended to create therapeutic programmes and interventions focusing on inducing, facilitating, and promoting hope and meaning in life as an effective way to achieve the subjective well-being of alcohol-addicted patients. This kind of solution could constitute an additional therapeutic offer addressed to alcohol-dependent patients.

## 6. Limitation and Future Research

The conducted study has some limitations.

Generalizability of the results is limited to alcohol-dependent individuals from Poland participating in AA. The research sample was homogeneous with regards to religious denomination. All participants had a Roman Catholic affiliation. The mean of abstinence duration for participants was relatively long, which means that they can be treated as addicts with stable abstinence.

The conducted study had a cross-sectional, not longitudinal, character. The cross-sectional model of research gives the possibility to interpret the direction of identified relationships between variables but not from the causation perspective. The alternative direction between research variables was not examined, which means that the potential role of subjective well-being as a consequence of involvement in AA and the antecedent of existential well-being should not be excluded. Additional research in the longitudinal model is necessary to investigate the relationships examined from the cause-and-effect perspective and confirm their directions.

The sample study was relatively small, and the bootstrapping method was used as a good solution in case of normally distributed variables. The small sample size was why it did not add to model-controlled variables.

The research encompassed relationships between latent variables, which are not observed variables. It means that results can be interpreted only as associations between latent variables, which could differ from connections between observed variables.

Because these findings must be interpreted as applying to Roman Catholics affiliated with AA from Poland, additional research is needed to investigate the confirmed model among representatives of another religious denominations, agnostics and atheists, alcoholics of other races, as well as in another cultural contexts. AA is a worldwide mutual self-help group, but it does not mean that there is a universal model of recovery due to involvement in self-help groups through existential well-being.

Future research should take into consideration religious facets of functioning, which are important predictors of participants’ meaning in life and hope. For more complexity, other results of involvement in AA, such as social support [34,37] or self-efficacy [36,37] should be incorporated. It is important to check which aspects of AA involvement are the strongest predictors of meaning in life and hope.

It would be interesting to verify the indirect impact of AA involvement on subjective well-being through hope and meaning in life using a bi-dimensional model of meaning in life [117]. Using other measures of well-being, quality of life and health—especially regarding not only psychological but additional social and physical spheres of life—could give interesting results.

In the future, research samples should be bigger and selected for alcohol-dependent individuals who have recently started their AA participation and do not yet experience a stable abstinence. It seems to be important to test whether this model is useful when the dependent variable is abstinence or frequency of alcohol use. A bigger sample could give an opportunity to verify if sex, religious involvement, other addictions, or diseases play a moderating role between involvement in AA and meaning of life/hope. In addition, it is important to verify whether the presented model of recovery could be employed to other self-help groups dedicated to both substance and behavioural addictions, such as drug addicts, gamblers, sex addicts, workaholics, etc.

## 7. Conclusions

The conducted study has confirmed the benefits resulting from involvement in AA. Engagement in this form of support of alcohol-dependent individuals from Poland was positively connected with existential well-being, which in turn positively related to subjective well-being.

## Figures and Tables

**Figure 1 ijerph-19-05173-f001:**
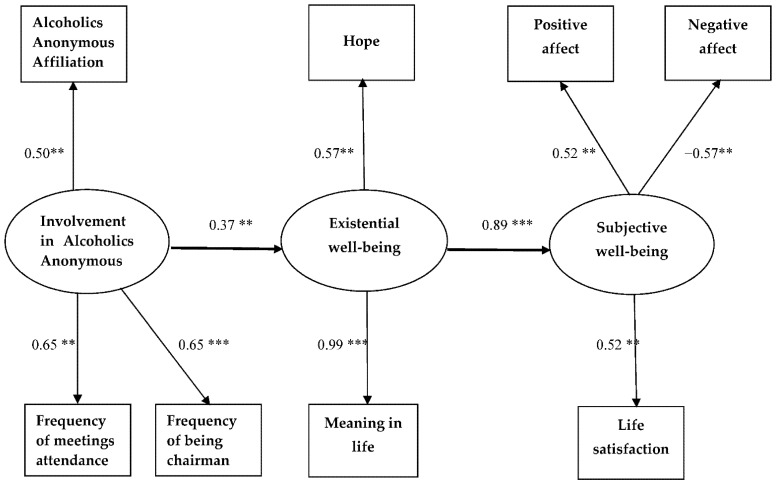
Path analysis results. Note: The standardized regression coefficients are presented. ** *p* < 0.01, *** *p* < 0.001. (Source: author’s research).

**Table 1 ijerph-19-05173-t001:** Demographics variables (*n* = 70).

	Classification	Percentage or Mean
Gender	Men	73.9%
	Women	26.1%
Age		46.1 years
Education	Elementary education	5.8
	Occupational education	29%
	High school education	46.4%
	University education	18.8%

**Table 2 ijerph-19-05173-t002:** Descriptive statistics (*n* = 70).

	PANAS Positive	PANAS Negative	PIL	HHI	AAI	Quantity of Steps	Frequency of AA Attendance	Frequency of Being Chairman	Abstinence Duration	Duration of AA Participation
Mean	21.47	21.37	108.14	38.54	3.77	6.96	100.31	7.97	76.04	102.76
Standard deviation	4.42	7.39	14.84	4.57	1.42	4.65	66.53	10.3	63.42	71.38
Skewness	−0.46	0.76	−0.88	−0.01	−0.53	−0.07	1.05	1.83	0.88	0.73
Kurtosis	0.65	0.03	0.96	−0.73	0.47	−1.78	1.06	3.15	−0.9	0.9
Minimum	7	5	66	29	0	0	5	0	1	1
Maximum	30	41	134	47	6	12	100	40	245	312
Reliability	0.90	0.78	0.79	0.80		-	-	-		

PANAS positive—positive affect from Positive and Negative Affect Schedule, PANAS negative—negative affect from Positive and Negative Affect Schedule, PIL—Purpose in Life Test, AAI—Alcoholics Anonymous Involvement Scale, HHI—Hertz Hope Index. (Source: author’s research).

**Table 3 ijerph-19-05173-t003:** Correlation matrix (*n* = 70).

	1	2	3	4	5	6	7
1. Life satisfaction							
2. Positive affect	0.35 **						
3. Negative affect	−0.24 *	−0.28 *					
4. Meaning in life	0.46 **	0.46 **	−0.52 **				
5. Hope	0.40 **	0.50 **	−0.33 **	0.57 **			
6. AA aliffiation	0.23	0.23 *	−0.24 *	0.36 **	0.26 *		
7. Frequency of AA attendance	0.07	0.09	−0.10	0.19	0.16	0.30 *	
8. Frequency of being chairman	0.16	0.01	−0.03	0.20	0.20	0.20 *	0.46 **

* *p* ≤ 0.05. ** *p* ≤ 0.01. (Source: author’s research).

## Data Availability

The data presented in this study are available on request from the corresponding author. The data are not publicly available due to privacy of the participants.

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
