# Peer review of "The Beneficial Role of Involvement in Alcoholics Anonymous for Existential and Subjective Well-Being of Alcohol-Dependent Individuals? The Model Verification"

_ijerph, 2022, doi:10.3390/ijerph19095173_

Round 1

Reviewer 1 Report

I am satisfied with the modifications

Author Response

Dear Academic Editor, thank you for your suggestions. I corrected every mistake you suggested and others I noticed.

This manuscript is a resubmission of an earlier submission. The following is a list of the peer review reports and author responses from that submission.

Round 1

Reviewer 1 Report

Understanding the mechanisms through which AA is beneficial is a critical area of research. I’m glad the study undertakes this important question.

While the author acknowledges in the limitations that the study is cross-sectional, I think more work needs to be done to discuss some of the limitations of this study.

Firstly, I would love for the author to more thoroughly establish the expected relationship between existential well-being and subjective well-being. How do we know they are distinct constructs? Why do we think existential well-being contributes to subjective well-being and not the other way around? Please more fully establish this reasoning in the introduction.

Secondly, were any additional control variables employed? You identified several plausible mechanisms through which AA benefits participants (e.g., motivation, belonging, etc). How do we know the relationship between AA involvement and subjective well-being isn’t due to those factors? If possible, please add some of these control variables to the model. If not, please more fully elaborate these issues in the limitations section.

P1 Line 12: Capitalize Polish

P1 Line 28: Alcohol dependent not dependence

P2 Line 55: Convincement is not the clearest word choice. Is there another word that more explicitly articulates what you mean to communicate?

P2 Line 61: I’m confused how the first sentence connects to the rest of the paragraph.

P5 Line 205- What about sex? Please clarify.

Author Response

Dear reviewer, thank you for your reasonable and accurate suggestions. I improved the text according to them.

Reviewer 2 Report

Dear Authors - congratulations on your interesting paper.  

The possibility that existential well-being (meaning and hope in life) can be treated as a parallel spiritual mechanism  (line 134) is an interesting proposition and seems to be backed up by some of the literature (Mac Donald - line 272). You have however not clearly defined the difference between religion and spirituality  - this might be relevant, especially as you suggest that existential well-being  (meaning and hope in life) can be treated as a parallel spiritual mechanism.  In various places you use the terms spiritual-religious practices or facets (111- 494) implying that the two terms are similar, this suggests that existential well-being meaning and hope in life) can be treated as a parallel spiritual/religious mechanism, rather than just a spiritual mechanism. Can they clarify your position?

I feel the method and results are clear and well presented as is your discussion and limitations. 

Author Response

Dear reviewer, thank you for your reasonable and accurate suggestions. I agree with your comments. I improved the text according to them.
